# MiR-30a-5p Alters Epidermal Terminal Differentiation during Aging by Regulating BNIP3L/NIX-Dependent Mitophagy

**DOI:** 10.3390/cells11050836

**Published:** 2022-02-28

**Authors:** Fabien P. Chevalier, Julie Rorteau, Sandra Ferraro, Lisa S. Martin, Alejandro Gonzalez-Torres, Aurore Berthier, Naima El Kholti, Jérôme Lamartine

**Affiliations:** 1CNRS UMR 5305, Tissue Biology and Therapeutic Engineering Laboratory (LBTI), 69007 Lyon, France; julie.rorteau@ibcp.fr (J.R.); ferraro.sandra2018@outlook.fr (S.F.); lisa.martin@ibcp.fr (L.S.M.); alejandro.gonzalez-torres@ibcp.fr (A.G.-T.); aurore.berthier@ibcp.fr (A.B.); nelkholti@ibcp.fr (N.E.K.); 2Claude Bernard University Lyon 1, 69100 Villeurbanne, France; 3Gattefossé SA, 36 Chemin de Genas, CS 70070, CEDEX, 69804 Saint-Priest, France

**Keywords:** BNIP3L, miR-30a, mitophagy, keratinocyte, aging, mitochondria

## Abstract

Chronological aging is characterized by an alteration in the genes’ regulatory network. In human skin, epidermal keratinocytes fail to differentiate properly with aging, leading to the weakening of the epidermal function. MiR-30a is particularly overexpressed with epidermal aging, but the downstream molecular mechanisms are still uncovered. The aim of this study was to decipher the effects of miR-30a overexpression in the human epidermis, with a focus on keratinocyte differentiation. We formally identified the mitophagy receptor BNIP3L as a direct target of miR-30a. Using a 3D organotypic model of reconstructed human epidermis overexpressing miR-30a, we observed a strong reduction in BNIP3L expression in the granular layer. In human epidermal sections of skin biopsies from donors of different ages, we observed a similar pattern of BNIP3L decreasing with aging. Moreover, human primary keratinocytes undergoing differentiation in vitro also showed a decreased expression of *BNIP3L* with age, together with a retention of mitochondria. Moreover, aging is associated with altered mitochondrial metabolism in primary keratinocytes, including decreased ATP-linked respiration. Thus, miR-30a is a negative regulator of programmed mitophagy during keratinocytes terminal differentiation, impairing epidermal homeostasis with aging.

## 1. Introduction

Aging is a natural and inexorable biological evolution associated with a progressive decline in tissue homeostasis. Skin is an excellent model of aging: during the natural aging process, the skin undergoes a typical age-related tissue dysfunction, including epidermis atrophy, barrier dysfunction and delayed wound healing. This chronological and dynamic process starts in the mid-20s and is easily remarkable at the macroscopic level at an advanced stage. However, early microscopic events such as changes in cell behaviour occur very early in the aging process and could be targeted to slow down the loss of tissue homeostasis. Although skin aging is strongly modulated by extrinsic factors, clues that a universal epigenetic program is driving the intrinsic tissue decline are converging in the literature. Indeed, multiple epigenetic changes are considered to be reliable hallmarks of tissue aging, such as modification of DNA methylation patterns, histone posttranslational modifications and modulation of non-coding RNA expression [1,2]. The latter is an emerging scientific domain, in which a number of studies have been published over the last decade. Indeed, the control of skin homeostasis largely depends on the fine-tuning of signalling pathways by long non-coding RNAs, circular non-coding RNAs and micro-RNAs (miRNAs) [3,4,5].

We previously conducted a genome-wide miRNA profiling in human primary keratinocytes from skin biopsies of young or elderly donors [6]. Thanks to this analysis, we identified miR-30a as the most over-expressed miRNA during the aging process in keratinocytes. The overexpression of miR-30 in young cells is sufficient to recapitulate the major functional defect of skin aging, namely, the disrupted barrier function in a 3D organotypic model [6]. This functional alteration in the epidermis is likely due to increased apoptosis, together with failure of the complete differentiation process. The epidermis is a multi-layered epithelium, mainly composed of keratinocytes. Keratinocytes from the basal layer are proliferative cells that continuously divide to self-renew the pool of progenitor cells and sustain the daily need for committed cells. These committed keratinocytes will undergo progressive differentiation, in stages, along a vertical axis that is oriented outwards. At the terminal stage of differentiation, keratinocytes can be likened to mummified cells, called corneocytes, devoid of nucleus and organelles.

Macroautophagy (hereafter referred as autophagy) plays a crucial role in the terminal differentiation of keratinocytes; therefore, its abnormal activity is associated with different cutaneous diseases [7]. Autophagy is a dynamic process that degrades and recycles cellular components, such as misfolded proteins and damaged organelles. Multiple molecular complexes are involved in autophagy, but they will all lead to the sequestration of the targeted component within a double-membrane autophagosome. This autophagosome will later fuse with a lysosome, leading to a degradative autolysosome [8]. In the epidermis, autophagy is constitutively active in the granular layer, where nuclei are cleared, leading to the terminal differentiation [9]. In addition, mitophagy, a specialized type of autophagy that targets mitochondria for elimination, is a key step to activate the terminal differentiation of keratinocytes, through the BNIP3/BNIP3L (also known as NIX) pathway [7,10,11]. Interestingly, autophagy is a recurrent biological pathway targeted by the miR-30 family in other systems, especially in many types of cancers [12,13,14]. In addition, other miR-30 family members, miR-30c and miR-30e, have previously been implicated in the targeting of BNIP3L in renal epithelial cells [15,16]. However, the functional role of miR-30a in skin, and the consequences of its overexpression with aging, are still uncovered in the literature. We hypothesized that miR-30a is a central regulator of mitophagy, which will disturb the normal process of epidermal differentiation with aging. The present study was conducted on a 3D organotypic model of reconstructed epidermis and on human skin biopsies. We formally identified a new miR-30a target and demonstrated that the mitochondrial dynamics are altered during differentiation along chronological aging.

## 2. Materials and Methods

### 2.1. Cell Culture

Human primary keratinocytes (HPK) were isolated in-house from skin biopsies as previously described [6], or purchased from Lonza (Basel, Switzerland, #00192627). Skin biopsies were obtained from the DermoBioTec tissue bank at Lyon (Tissue Transfer Agreement n°214854) with the informed consent of adult donors or parents of children undergoing surgical discard (non-pathological tissues from foreskin, ear, breast, or abdomen), in accordance with the ethical guidelines (French Bioethics law of 2021). Donor specifications are indicated in Appendix A. HPK were cultured in KGM Gold medium (Lonza, #00192060) at 37 °C and 5% CO_2_. The culture medium was renewed three times a week and cells were maintained to no more than passage 4. The differentiation of HPK was induced by leaving the cells at confluency in KGM Gold medium. Human Embryonic Kidney (HEK293T) cells were cultured in Dulbecco’s modified Eagle’s Medium–Glutamax medium (DMEM) (Thermo Fisher Scientific, Villebon-sur-Yvette, France, #6196-026) 10% FBS at 37 °C and 5% CO_2_, with renewal of culture medium three times a week.

### 2.2. Transfection and Luciferase Assay

HPK at 70% confluency were transfected with miRNA mimics at 20 nM: miR-30a-5p (Thermo Fisher Scientific, #4464066, assay ID: MC11062), miR-30a-3p (Thermo Fisher Scientific, #4464066, assay ID: MC10611) and miRNA negative control (Thermo Fisher Scientific, #4464058), using RNAiMax (Thermo Fisher Scientific, #13778075) according to the manufacturer’s instructions. The cells were processed for RT-qPCR analysis of *BNIP3L* expression 24 h after transient transfection. Partial sequence of *BNIP3L* 3’UTR was cloned downstream of a firefly luciferase reporter plasmid (VectorBuilder, Neu-Isenburg, Germany, #VB191204-1713qzf). The plasmid was then submitted to site-directed mutagenesis using the In-Fusion technology (Takara Bio Europe, Saint-Germain-en-Laye, France, #638909) with inverse PCR (see Appendix A) according to manufacturer’s instructions. The characterization of the resultant plasmids was performed by Sanger sequencing (Biofidal, Vaulx-en-Velin, France). An additional renilla luciferase reporter plasmid was generated to serve as an internal standardizer (VectorBuilder, #VB191205-1069fzu). HEK293T cells at 70% confluency were co-transfected with the firefly luciferase reporter plasmids at 400 ng/mL, the renilla luciferase reporter at 0.4 ng/mL and miRNA mimics at 20 nM using TransIT-X2 (Mirus Bio, Madison, WI, USA, #MIR 6000) according to manufacturer’s instructions. Eighteen hours after transfection, the luciferase activity was measured using the renilla-firefly luciferase dual assay kit (Thermo Fisher Scientific, #16186) in a microplate reader (Infinite M1000, Tecan, Männedorf, Switzerland).

### 2.3. Protein Extraction and Immunoblotting

Total proteins were extracted using RIPA buffer (50 mM Tris-HCl pH = 8, 150 mM NaCl, 1.5 mM KCl, 1% NP-40, 0.1% SDS, 0.5% sodium deoxycholate, 0.1% Triton X-100, 1 mM EDTA) containing a protease and phosphatase inhibitors cocktail (Thermo Fisher Scientific, #A32961). After quantification with the Pierce BCA Protein Assay Kit (Thermo Fisher Scientific, #23225), 20 µg of proteins were loaded on an 10% SDS-polyacrylamide gel and transferred onto a PVDF membrane (Merck, Darmstadt, Germany, #IPVH85R). The membrane was incubated for 1 h at RT in blocking buffer (TBS-Tween-20 0.1%, 5% BSA) and immunoblotted overnight at 4 °C with primary antibodies targeting BNIP3L or Vinculin. After washing, HRP-conjugated secondary antibodies were incubated for 1 h at RT (see Appendix A for antibody details). Proteins were revealed using SuperSignal West Pico PLUS Chemiluminescent Substrate (Thermo Fisher Scientific, #34580) and the signal was detected by the Fusion Fx system (Vilber, Marne-la-Vallée, France).

### 2.4. Reconstructed Human Epidermis (RHE) Production

RHE were prepared as described previously in Muther et al., 2017 [6]. In brief, 3 × 10^4^ primary fibroblasts were seeded on the outer face of polycarbonate membrane of cell culture inserts (Sigma-Aldrich, Saint-Quentin-Fallavier, France) and cultured for two days. Then, 3 × 10^5^ primary keratinocytes, infected by pSLIK Venus control or pSLIK Venus miR-30a, were seeded on the inner face of inserts. Three days after, differentiation and stratification were induced when keratinocytes were placed at the air–liquid interface. The culture was maintained for 11 days, and culture medium was changed every day during the immersion phase. Doxycycline (Sigma-Aldrich) at 0.1 μg/mL was added for the duration of the protocol to induce miR-30 over-expression.

### 2.5. Immunofluorescence

Human paraffin embedded tissue microarray (TMA: #SK2444A and #SKN1001, see Appendix A for details) were purchased from US Biomax (Derwood, MD, USA). Antigen retrieval (10 mM sodium citrate buffer, 0.05% Tween20) was performed at 95 °C for 20 min before immunostaining. RHE were prepared for immunofluorescence as previously described [6]. The following steps are similar for both TMA and RHE sections. After dewaxing and rehydration, tissue sections were permeabilized using PBS 0.1% Triton X-100, 0.1 M glycine during 10 min at room temperature (RT). Samples were then blocked with PBS containing 5% goat serum, 2% BSA and 0.1% Tween20 for at least 1 h at RT. After washing steps, primary antibodies were incubated overnight at 4 °C (Appendix A). After washings, secondary antibodies were incubated 45 min at RT and nuclear staining was performed using ProLong^TM^ Glass Antifade Mountant with NucBlue^TM^ (Thermo Fischer Scientific, #P36981). Negative controls were performed by omitting primary antibodies. Images were visualized using High Content Screening Yokogawa CQ1 microscope (Yokogawa, Tokyo, Japan), digitalized using sCMOS camera (Olympus, Hamburg, Germany) and analysed using ImageJ software (version 2.1.0/1.53c).

### 2.6. Total RNA/DNA Isolation and Real-Time Quantitative PCR

Total RNA and DNA were isolated using Quick-DNA/RNA^TM^ Miniprep kit (Zymo research, Mülhauser, Germany) according to manufacturer’s instruction. First, mRNA were reverse-transcribed into cDNA using PrimeScript^TM^ RT reagent kit (Takara Bio Europe, #RR037A) and analysed in real-time qPCR using SYBR^®^ Premix ExTaqII (Takara Bio Europe, # RR820A) on an AriaMx Realtime PCR system (Agilent Genomics, Santa Clara, CA, USA). Results were normalized to *TBP* and *RPL13A* housekeeping gene expression levels, using the 2^−ΔΔCt^ quantification method. Secondly, relative mitochondrial DNA content was calculated as the mean ratio of two mitochondrial genes copy number (*ND1* and *TL1*) to single-copy nuclear genes (*HBB* and *SERPINA1*) using the 2^−ΔΔCt^ quantification method. The detailed PCR program is indicated in the Appendix A and Methods and all the primers are listed in Appendix A.

### 2.7. Seahorse Analysis

Oxygen consumption rate (OCR) was measured with a Seahorse XF extracellular flux analyzer according to the manufacturer’s instructions (Agilent). Primary keratinocytes were seeded at 60,000 cells per well (four wells as technical replicate/cell type) in a 24-well cell culture microplate and incubated overnight at 37 °C in 5% CO_2_. Culture medium was replaced with XF assay medium supplemented with 10 mM glucose (Agilent, #103577-100), 1 mM pyruvate (Agilent, #103578-100) and 2 mM glutamine (Agilent, #103579-100, Agilent) and cells were incubated in the absence of CO_2_ for 45 min before measurement. OCR was determined before the injection of specific metabolic inhibitors and after successively adding 1.5 μM oligomycin, 1 μM FCCP, and 0.5 µM rotenone/antimycin A (Sigma-Aldrich). Wave software was used to analyse seahorse measurements.

### 2.8. Statistical Analysis

Data are expressed as mean or median ± SD. Statistical significance was calculated by Student’s *t*-test, one-way analysis of variance (ANOVA), two-way analysis of variance (ANOVA2), or Pearson correlation using Prism software (version 7.0, GraphPad Software, San Diego, CA, USA). Mean differences were considered statistically significant when *p* < 0.05. * *p* < 0.05, ** *p* < 0.01, *** *p* < 0.001, **** *p* < 0.0001.

## 3. Results

### 3.1. BNIP3L Is a New Identified Target of miR-30a-5p

As miR-30a is already well known to regulate autophagy through targeting *BECLIN1* or *ATG5,* mostly in cancer cells [12,13,14], we chose to focus on putative targets that were more specifically involved in mitophagy. Using TargetScan (http://www.targetscan.org/vert_80/, release 8.0: September 2021), we found that miR-30a-5p has two highly conserved putative binding sites in *BNIP3L* 3’UTR at positions 2350–2357 and 2651–2657 (Figure 1A). In addition, another putative binding site of miR-30a-5p (position 1126–1132) and two putative binding sites of miR-30a-3p (positions 1082–1088 and 2059–2065) were also found, even though the conservation of these sites among vertebrates is less important. Since both strands of miR-30a are overexpressed in keratinocytes from aged skin, we tested the in vitro ability of miRNA mimics to decrease the mRNA levels of *BNIP3L* in keratinocytes (Figure 1B)., Both mimics of miR-30a-3p and miR-30a-5p induced a significant decrease in *BNIP3L* mRNA levels, by 28% and 36%, respectively, 24 h after the transient transfection of proliferating human keratinocytes.

To further confirm the direct targeting of *BNIP3L* by the miR-30a, a portion of its 3’UTR sequence containing the five putative binding sites was cloned into a luciferase reporter plasmid. In combination with transient transfection of either miR-30a-3p or miR-30a-5p mimics in HEK293T cells, we observed a strong reduction (−40%) in the luminescence signal only in presence of the miR-30a-5p strand (Figure 1C). Thus, it appears that only miR-30a-5p directly targets the *BNIP3L* 3’UTR and that the effect of miR-30a-3p that was previously observed in primary keratinocytes is likely related to a secondary event. We then selectively and individually mutated the three putative binding sites of miR-30a-5p. Sanger sequencing showed a partial deletion of four of the seven nucleotides in the first binding site (1126–1132) and the complete deletion of all seven nucleotides from the two other binding sites (2350–2357 and 2651–2657) (Figure 1D). Mutants 1 and 3 partially rescued the luciferase activity (from −40% to −17% and from −40% to −19%, respectively), suggesting that these two sites only account for a partial activity of miR-30a-5p on *BNIP3L* 3’UTR. Surprisingly, the luciferase activity was almost fully restored (from −40% to −3%) when the intermediate binding site was mutated, suggesting that this single site is sufficient for miR-30a-5p activity. These apparently conflicting data may reflect non-canonical miRNA-target interactions, independently of the seed complementary sequence [17], in the proximity of the both sites 1126–1132 and 2651–2657.

Finally, we confirmed the functional targeting of *BNIP3L* by miR-30a-5p by Western blot analysis of the protein content. Indeed, after transient transfection of miR-30a-5p in primary keratinocytes, we found that the protein levels of BNIP3L were decreased by about 40% (Figure 1E,F), which corresponds to the level of mRNA decrease in the same conditions. Thus, miR-30a-5p is likely regulating the protein level of BNIP3L through mRNA degradation.

### 3.2. MiR-30a-5p Abolishes BNIP3L Expression in the Granular Layer of RHE

To confirm the regulation of *BNIP3L* by miR-30a in a more relevant system, we took advantage of an organotypic model of RHE generated from HPK overexpressing miR-30a by stable lentiviral transduction. Using three different human donors, we showed that keratinocytes normally express BNI3PL in the granular layer of the epidermis (Figure 2A), in accordance with its described function in the degradation of mitochondria during the cornification [11]. Strikingly, the overexpression of miR-30a in keratinocytes induced a strong reduction in BNIP3L staining in the granular layer. The reminiscence of BNIP3L protein is likely due to cell heterogeneity in the overexpression of miR-30a, since we used primary human keratinocytes that did not undergo selection pressure. Nevertheless, we measured the percentage of the granular layer length with positive staining for BNIP3L with or without miR-30a over-expression. Under normal conditions, 88% of the granular layer is positive for BNIP3L, whereas, in the presence of high levels of miR-30a, only 18.5% is stained for this mitophagy protein (Figure 2B).

### 3.3. BNIP3L Expression Decreases with Chronological Aging in Human Epidermis

Since miR-30a expression is known to increase with chronological aging in the epidermis, we used a TMA to screen the expression of its target BNIP3L in human skin biopsies of different ages. We defined two groups based on the age of the donor and on the structure of the skin: adult individuals (*n* = 11, from 19 to 42 years old, mean age at 34.4) vs. aged individuals (*n* = 10, from 61 to 78 years old, mean age at 68.3). Indeed, skin aging is characterized in terms of tissue morphology by epidermal atrophy and flattening of the dermo-epidermal junction. Illustrative examples of skin biopsies from three adults (Figure 3A, top panel: 38, 35 and 42 years old) and three aged individuals (Figure 3A, bottom panel: 61, 71 and 78 years old) show the pattern of BNIP3L staining together with KRT14, a specific marker of basal undifferentiated keratinocytes. We found that BNIP3L is highly expressed in the upper layers of epidermis in the adult group. In addition, BNIP3L staining is often delimitating cornified cells that are devoid of a nucleus (Figure 3A, inserts). With aging, we observed a strong decline in the BNIP3L signal, along with a substantial decline in the number of enucleated keratinocytes. We evaluated the correlation between the percentage of positive cells for BNIP3L staining and age and the correlation of the normalized signal intensity per epidermal area with age (Figure 3B,C). Using these two correlations, we found equivalent Pearson coefficients with r = −0.6542 (R^2^ = 0.428, *p* = 0.0024) and r = −0.6198 (R^2^ = 0.3841, *p* = 0.0036), respectively. These mathematical simulations strongly suggest that the chronological aging negatively affects the expression of BNIP3L in human epidermis.

### 3.4. Aging Is Associated with Alterations of Keratinocyte Terminal Differentiation, BNIP3L Expression and Mitochondrial Elimination

To confirm that BNIP3L is differently regulated with aging during epidermis differentiation, we used an in vitro 2D model of differentiation with primary keratinocytes isolated from three age cohorts: young individuals (*n* = 6, from 3 to 10 years old, mean age at 5), adult individuals (*n* = 6, from 26 to 46 years old, mean age at 36.5) and aged individuals (*n* = 5, from 68 to 92 years old, mean age at 79). Considering the young cohort as a reference, we examined RNA expression levels of several markers along a nine-day differentiation kinetic (Figure 4A).

The keratinocytes’ commitment to differentiation, illustrated by the expression of the early marker *KRT10*, was not found to be affected between young and adult groups, whereas keratinocytes from the aged group reduced *KRT10* expression by approximately 5-fold. With the progression of differentiation, we noticed that the expression levels of most markers (*IVL*, *LOR*, *TGM1*, *FLG*) in the adult group were not very different from those in the young group. One exception was the expression of *LOR,* which dropped at day 9 in the adult group, while it was maintained in the young group. Considering the keratinocytes in the aged group, we observed both a delay in the intensity of expression and a strong decrease in the maximal level of expression (*IVL*, *LOR*, *FLG*), except for *TGM1,* which is homogeneously expressed within the three age groups. However, the expression of the terminal differentiation genes, *KLK7*, *AQP9* and *CDSN*, were significantly decreased in both the adult group and the aged group, as compared to the young one. Interestingly, we found that *BNIP3L* was normally overexpressed from day 6 of differentiation, a time corresponding to the switch from late differentiation to terminal differentiation, as illustrated by the concurrent increase in *KLK7*, *AQP9* and *CDSN*. The pattern of *BNIP3L* expression during differentiation in keratinocytes summarized the profile of the different age groups. The maximal expression level was reached from day 6 and maintained at day 9 in keratinocytes from young donors. In cells from adult donors, we observed a delayed maximal expression level at day 9, but with a similar intensity of expression to the young group. Finally, keratinocytes from aged donors also expressed *BNIP3L* at the maximal at day 9, but shiwed a significant reduction in intensity compared with young and adult groups. This result is in accordance with the variation in BNIP3L immunostaining during aging in the human epidermis sections. We summarized the moment and the intensity of the maximal expression levels (relative to D0) for each gene in each group in Table 1.

Considering the role of BNIP3L in mitochondrial fragmentation associated with epidermis terminal differentiation [11], we then evaluated the relative amount of mitochondria in the three age cohorts during the 2D differentiation kinetic. Remarkably, we observed a strong inter-individual variation in the number of mitochondria per proliferating keratinocytes (D0), with no difference in the overall median between the age groups (Figure 4B). In the young group, the number of mitochondria progressively increased, reaching a peak at day 6, and then dropped at day 9 (Figure 4C; relative to D0: D2 = 80%, D4 = 100%, D6 = 135%, D9 = 90%). We believe that the increase in mitochondria is necessary during differentiation to keep up with the high metabolic demand before the terminal differentiation. In 2D, there was no complete elimination of the organelles and nucleus because the final cornification step requires contact with air. In the adult group, we observed an early increase in the mitochondria content, as soon as day 4, which was maintained at day 6 and even pursued at day 9 (relative to D0: D2 = 110%, D4 = 135%, D6 = 125%, D9 = 260%). The evolution of mitochondria in keratinocytes from the aged group followed the same profile as those of the adult group, with a milder intensity but with a retention of mitochondria at day 9 as well (relative to D0: D2 = 120%, D4 = 107%, D6 = 100%, D9 = 241%). These results overlap with the alteration in BNIP3L expression from day 6 in the adult and aged groups and may reflect a defective mitophagy during terminal differentiation of keratinocyte. However, the increased expression of BNIP3L at day 9 in the adult group suggests that either the induction of BNIP3L was too late for proper mitochondrial elimination, or that additional mitochondrial dynamic mechanisms are also involved.

### 3.5. Aged Keratinocytes Display Mitochondrial Metabolic Defects

As mitophagy serves as a quality control process of functional mitochondria, compromised mitophagy may alter the mitochondrial metabolism. Therefore, we measured the OCR in primary keratinocytes from the three age groups and compared the effect of miR-30a overexpression on the same parameters in parallel. We found that the basal OCR was significantly decreased in the adult group (−22.7%) and the aged group (−32.1%), as compared to the young one (Figure 5A). Similarly, the overexpression of miR-30a in young cells also induced a 29.3% decrease in the basal respiration. Using the ATP-synthase inhibitor oligomycin, we found that ATP-linked respiration was also significantly decreased, by 23.5% in the adult group, 30.6% in the aged group and 31.9% in the miR-30a overexpressing cells (Figure 5B). Next, we added FCCP, a protonophore which collapses the proton gradient across the mitochondrial inner membrane. This drug forces the electron transport chain to function at its maximal rate. Even if the maximal respiration capacity was altered identically to the basal respiration rate in the adult or the aged group (Figure 5C), the reserve capacity, expressed as the percentage of the basal rate, was identical within the three groups (Figure 5D). The overexpression of miR-30a also mimicked the difference between young and old groups, with a diminution of 54% of the maximal respiration. Lastly, to completely shut down the electron transport chain function, we added antimycin A and rotenone, two inhibitors of complex III and I, respectively. Consequently, the remaining OCR was associated to non-mitochondrial respiration and allowed for the determination of the proton leak during mitochondrial respiration. The non-mitochondrial oxygen consumption, related to the activity of diverse desaturase and detoxification enzymes, also seemed altered with aging. In the adult group, we observed a 30% decrease, albeit not significant (*p* = 0.0995), whereas in the aged group, this non-mitochondrial use of oxygen was decreased by half. In cells overexpressing miR-30a, we found a 33.7% decrease (Figure 5E). Finally, regarding other parameters, the proton leak decreased with aging or with miR-30a overexpression, but invariably accounted for about 20–25% of the basal respiration in all groups (Figure 5F). This result suggests that the proton permeability of the inner mitochondrial membrane was not affected by age. Of note, we observed a divergent profile of young keratinocytes transduced with the scrambled sequence as compared to unmodified young keratinocytes. Since the lentiviral transgene was under the control of an inducible element, we treated this with doxycycline to induce the overexpression. Doxycycline is known to alter mitochondrial metabolism [18], and this side effect is likely to explain the fact that transduced young keratinocytes with the scrambled sequence displayed a mitochondrial metabolism profile that was closer to the adult group than to the young group. However, the overexpression of miR-30a under the same conditions altered the mitochondrial metabolism to the same extent as the effect of chronological aging on primary cells. Thus, considering that the number of mitochondria did not vary with age (Figure 4B), it seems that the mitochondrial metabolic activity decreases with aging and miR-30a overexpression, which likely results in a decreased ATP synthesis.

## 4. Discussion

BNIP3L (BCL2 Interacting Protein 3 Like)/NIX is a mitochondrial outer membrane protein of the BH3-only member of the BCL2 family, initially identified as a pro-apoptotic inducer [19,20]. In addition to its role in the regulation of mitochondrial-dependent apoptosis, BNIP3L has also been identified as a critical autophagy receptor for the selective clearance of mitochondria in mammalian cells. For instance, BNIP3L is essential for mitochondria removal during the terminal differentiation of multiple cell types, including erythroid cells [21,22], ocular lens fiber cells [23], cardiac progenitor cells [24] and optic nerve oligodendrocytes [25]. A recent report showed that BNIP3L is also critical to the terminal differentiation of epidermal keratinocyte [11]. Indeed, the authors used CRIPSR-Cas9 to knock-out BNIP3L in immortalized human keratinocytes (hTERT cell line) and generated reconstructed epidermis from these modified cells. They obtained a stratified epithelium but noticed maturation defects. In details, they described that BNIP3L-deficient keratinocytes failed to undergo through complete cornification, with retention of mitochondria in the uppermost layers. In the present study, we found that BNIP3L is a direct target of miR-30a and we have previously reported that miR-30a expression levels substantially increase with aging in human keratinocytes [6]. Skin aging is associated with epidermal differentiation defects [26,27], and we previously demonstrated that overexpression of miR-30a alters the differentiation process of keratinocytes. Indeed, using an organotypic model of reconstructed human epidermis (RHE), we have demonstrated that miR-30a overexpression strongly downregulates the differentiation markers such as KRT1, KRT10, IVL and LOR, and decreases the efficacy of the barrier function [6]. Accordingly, we found here that BNIP3L expression is strongly reduced with aging in human skin. Using a 2D model of keratinocyte differentiation, we also observed a retention of mitochondria with aging, thus imitating the findings of the study with BNIP3L-deficient keratinocytes [11]. In addition, a previous report also showed that the overexpression of miR-30e-5p, a closely related micro-RNA to miR-30a-5p, is associated with an increased number of mitochondria in podocytes, as well as through BNIP3L downregulation [15]. In addition to the reduction in BNIP3L expression with age and its role in the differentiation process, we also found that the mitochondrial metabolism is affected during chronological aging.

Mitochondrial metabolism is the main source of cellular energy, generating high levels of ATP. In addition to ATP’s role providing energy for many cellular processes, it has been demonstrated that the inhibition of ATP synthase blocks the differentiation of keratinocytes [28]. In this study, exposing the HaCaT cell line to oligomycin induced a significant decrease in the intracellular ATP content while extracellular ATP increased in parallel. While no effect on keratinocytes proliferation was noted, the decreased intracellular ATP concentration was associated with an inhibitory effect on involucrin expression. Using a confluence-dependent differentiation model, as we used in the present study, the authors also showed that the ATP5B subunit of ATP-synthase was induced at days 6 and 8 of differentiation, at both the mRNA and protein levels. Even if the molecular mechanisms are not yet elucidated, these results strongly suggest that the intracellular ATP produced by mitochondrial metabolism is closely related to the terminal differentiation of keratinocytes. Accordingly, we observed a significant reduction in the oxygen consumption associated with ATP synthesis in keratinocytes from aged donors, together with alterations in their terminal differentiation.

Interestingly, a recent study has pointed out that mitophagy plays a key role in the regulation of bioenergetics in neurons. Using a mouse model of Alzheimer’s disease (AD), they observed a deficiency in energetic enhancement upon oxidative phosphorylation stimulation in brains [29]. In their model, the high energetic demand normally induces mitochondrial turnover by mitophagy, which is impaired in AD. Since the onset of AD is strongly associated with aging, we can hypothesize that the observed deficit in oxidative phosphorylation in keratinocytes is also linked to defective mitophagy with aging. In the skin, another study has found that ATP content was higher in primary dermal fibroblasts from centenarians, as compared to fibroblasts isolated from adult (about 27 years old) and aged (about 75 years old) human subjects, even though centenarians also presented oxidative phosphorylation defects [30]. The preservation of an adequate ATP production was associated with an increased mitochondrial mass, together with re-arrangement of the mitochondrial network. Mitochondrial dynamics, a balance between mitochondrial fission and fusion, is critical to sustain functional mitochondria. Fission allows for the creation of new mitochondria, but it also contributes to quality control by removing the damaged components. Conversely, fusion relieves stress by mixing the damaged and undamaged mitochondria contents. In dermal fibroblasts from centenarians, the authors found more hyperfused and elongated mitochondria compared to younger subjects. They concluded that mitochondrial fission was reduced in individuals undergoing what we might call “successful aging”. Since our study did not include cells from centenarians, it would be interesting to verify whether the remodelling of the mitochondrial network observed in dermal fibroblasts also occurs in keratinocytes from long-living individuals.

Finally, even if changes in mitochondrial dynamics have been shown to contribute to other age-related disorders, such as cardiac disorders, neurodegenerative diseases and sarcopenia (reviewed in Reference [31]), mitophagy and the balance between fission and fusion are not the only processes that can be used to achieve mitochondrial quality control. In response to oxidative stress, cells initiate an external release of mitochondria [32]. This mechanism occurs as an early response and precedes mitophagy in HeLa cells. To ensure cellular homeostasis, damaged mitochondria or their constituents will be transferred out of the cells in free form or encapsulated in extracellular vesicles (EVs). This second process functions in complementarity with mitophagy: it is repressed when mitophagy is efficient, whereas it is activated to compensate for a lack of mitophagy [33]. In the present study, we demonstrated that aged cells, which highly express miR-30a, exhibit a defective BNIP3L-dependent mitophagy in the last steps of keratinocyte differentiation. In an aging context, the external mitochondrial release by keratinocytes does not seem to be enough to compensate defective mitophagy and restore mitochondrial homeostasis. This statement is in good agreement with a previous study, which reported that the amount of mitochondrial DNA released in EVs decreases with age [34]. Thus, the mitochondrial dysfunction observed in skin aging may be due to breakdown in the two mitochondrial quality control systems. Moreover, EVs, as key vectors of intercellular communication [35], allows for the transfer of intact mitochondria or functional mitochondrial proteins and DNA to other cells. Several studies have analysed the effect of EVs containing mitochondrial components on other cells [36], and it has notably been shown that EVs from older individuals affect the mitochondrial function of HeLa cells with a significant decrease in basal respiration [34]. Interestingly, we found in this study that keratinocytes from adults have an intermediate profile. They had higher expression levels of BNIP3L than keratinocytes from aged individuals, but they already displayed the equivalent altered mitochondrial metabolic activity, including lower basal respiration, as compared to cells from young individuals. Thus, one can hypothesize that the extracellular release of mitochondria plays an important role in the propagation of aging signals in skin tissue, leading from an adult phenotype to an aged one. Additional studies are needed to further validate this assumption.

## Figures and Tables

**Figure 1 cells-11-00836-f001:**
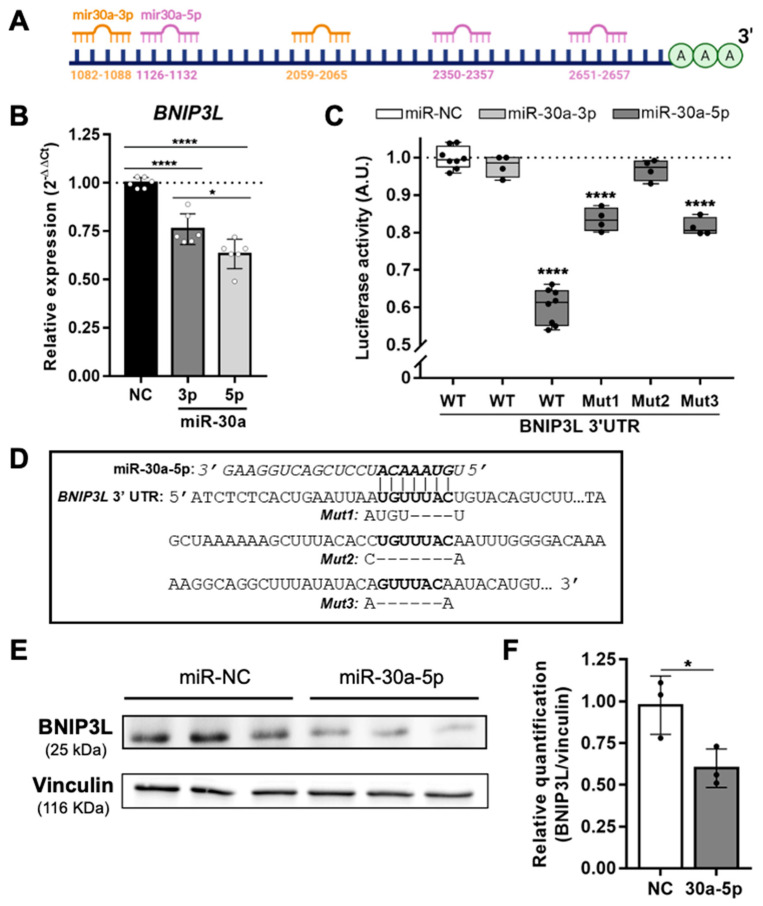
miR-30a interactions with *BNIP3L*. (**A**) Schematic of the *BNIP3L* 3’UTR region showing the putative miR-30a binding sites. (**B**) Relative mRNA expression of *BNIP3L* in keratinocytes 24 h after transfection of miR-30a-3p or miR-30a-5p mimics or a scrambled sequence (NC: Negative Control). qPCR analysis was normalized to *TBP* and *RPL13A* housekeeping genes using the 2^−ΔΔCt^ quantification method (mean ± SD, *n* = 6; * *p* < 0.05 and **** *p* < 0.0001). Exact *p*-values were determined using the one-way ANOVA and Tukey post-hoc tests. (**C**) Normalized luciferase activity (red firefly/green renilla) after co-transfections of HEK293T cells with the Wild-Type (WT) or Mutant (Mut) reporter plasmids and miRNA mimics (median ± SD, *n* = 4 or *n* = 8; **** *p* < 0.0001). Exact *p*-values were determined using the one-way ANOVA and Tukey post-hoc tests. (**D**) Schematic of a portion of *BNIP3L* 3’UTR region with the three putative complementary pairing sites with miR-30a-5p. The mutated sites, obtained by inverse PCR and characterized by Sanger sequencing, are also indicated below the original sequence. (**E**) Western blot analysis of keratinocytes of BNIP3L (25 kDa) expression 36 h after transfection of miR-30a-5p mimics or a scrambled sequence (NC: Negative Control). Western blot analysis of Vinculin (116 kDa) was used as a loading normalizer. (**F**) Quantification of BNIP3L expression normalized by the expression of Vinculin (mean ± SD, *n* = 3; * *p* < 0.05). Exact *p*-value was determined using the Student’s *t*-test.

**Figure 2 cells-11-00836-f002:**
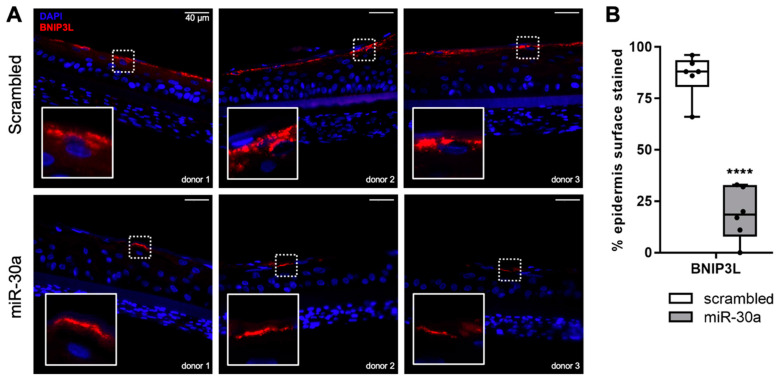
Effect of miR-30a on BNIP3L expression in the granular layer of reconstructed epidermis. (**A**) Immunofluorescent staining of BNIP3L in reconstructed human epidermis overexpressing miR-30a or a scrambled sequence by stable lentiviral transduction. Representative photographs of 3 independent replicates are shown. (**B**) Quantification of the surface of the epidermis with a positive signal for BNIP3L in each condition (median ± SD, *n* = 6; **** *p* < 0.0001). Exact *p*-value was determined using the Student’s *t*-test.

**Figure 3 cells-11-00836-f003:**
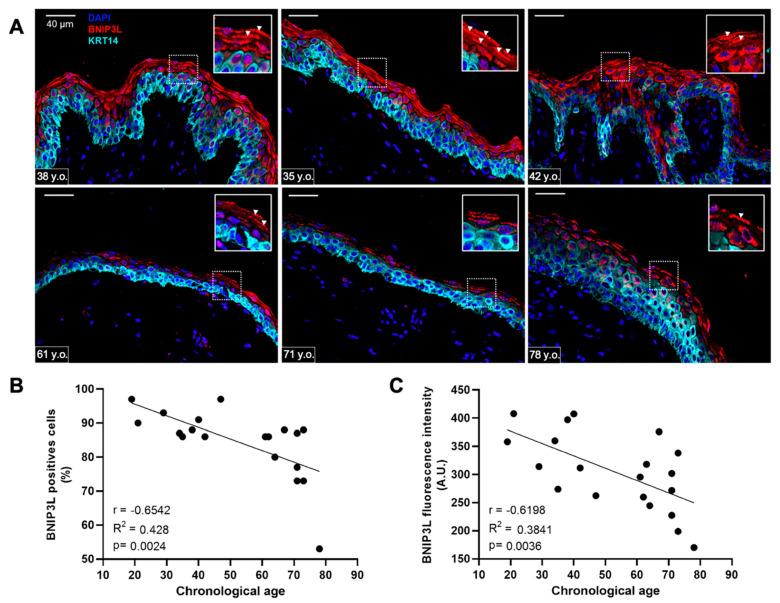
BNIP3L expression in human skin biopsies from individuals at different ages. (**A**) Immunofluorescent staining of BNIP3L and KRT14 in human skin sections. Representative photographs of 6 individuals from two different age groups: adult, or aged people (top, from left to right: 38, 35 and 42 years old; bottom, from left to right: 61, 71 and 78 years old). Inserts are a magnification of the selected area showing BNIP3L-positive cells (white arrowheads) (**B**) Mathematical correlation between the percentage of BNIP3L-positive cells from the whole epidermis and the chronological age of the individual from which the skin biopsy has been sampled (*n* = 21). Exact *p*-value was determined using the Pearson correlation test. (**C**) Mathematical correlation between the total BNIP3L fluorescence intensity from the whole epidermis and the chronological age of the individual from which the skin biopsy has been sampled (*n* = 21). Exact *p*-value was determined using the Pearson correlation test.

**Figure 4 cells-11-00836-f004:**
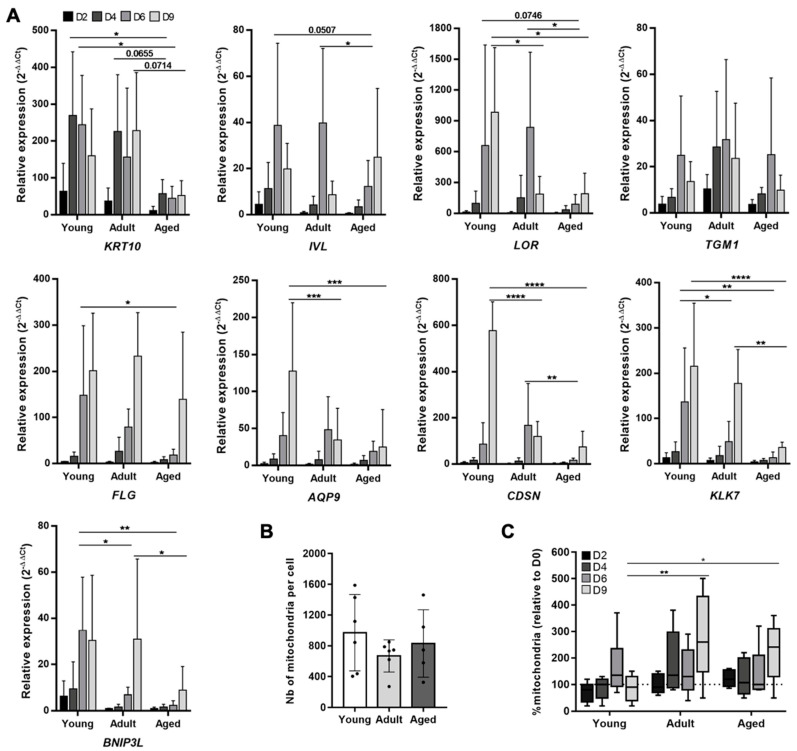
Human keratinocyte differentiation, *BNIP3L* expression and mitochondria retention in vitro with aging. (**A**) Relative mRNA expression of differentiation markers and *BNIP3L* in keratinocytes during a kinetic of differentiation in 2D culture. Three different age groups were tested (young group: *n* = 6, from 3 to 10 years old, mean age at 5; adult group: *n* = 6, from 26 to 46 years old, mean age at 36.5; aged group: *n* = 5, from 68 to 92 years old, mean age at 79) and followed over the time (0, 2, 4, 6 and 9 days after reaching confluency). qPCR analysis was normalized to *TBP* and *RPL13A* housekeeping genes (mean ± SD; * *p* < 0.05, ** *p* < 0.01, *** *p* < 0.001 and **** *p* < 0.0001). Exact *p*-values were determined using the two-way ANOVA and Tukey post-hoc tests. (**B**) Mitochondria content per cell estimated by qPCR at day 0. qPCR analysis was performed by normalizing the relative content of mitochondrial genes (*ND1* and *TL1*) with the content of nuclear genes (*HBB* and *SERPINA1*) using the 2^−ΔΔCt^ quantification method in the three groups (mean ± SD; young group: *n* = 6; adult group: *n* = 6; aged group: *n* = 5). (**C**) Evolution of the content of mitochondria expressed as a percentage of the quantity at day 0 (median ± SD; * *p* < 0.05, ** *p* < 0.01). Exact *p*-values were determined using the two-way ANOVA and Tukey post-hoc tests.

**Figure 5 cells-11-00836-f005:**
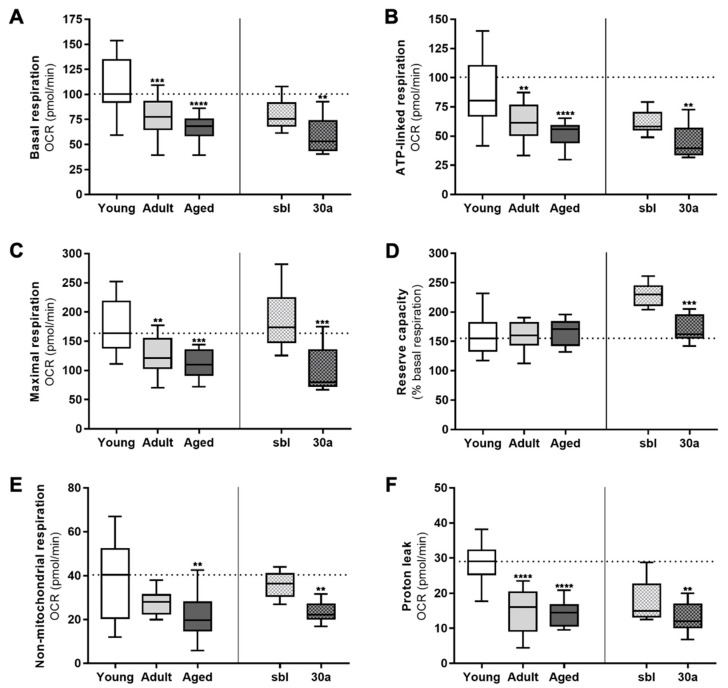
Mitochondrial metabolic activity of keratinocytes from different age groups and after overexpression of miR-30a. Major aspects of mitochondrial coupling and respiratory control were measured using the Seahorse Bioanalyzer and determined by the sequential additions of oligomycin, an ATP synthase inhibitor, FCCP, a protonophoric uncoupler, and rotenone and antimycin A, two inhibitors of the electron transport chain. sbl: miR-scrambled; 30a: miR-30a. Basal respiration (**A**), ATP-linked respiration (**B**), maximal respiration (**C**), reserve capacity (**D**), non-mitochondrial respiration (**E**) and proton leak (**F**) were determined by measuring the Oxygen Consumption Rate (OCR) in the culture media (median ± SD; young group: *n* = 6; adult group: *n* = 6; aged group: *n* = 5; sbl group: *n* = 3; 30a group: *n* = 3; ** *p* < 0.01, *** *p* < 0.001 and **** *p* < 0.0001). Exact *p*-values were determined using the one-way ANOVA and Tukey post-hoc tests for comparison between ages and paired Student’s *t*-test for comparison between cells transfected with either scramble miRNA sequence or miR-30a.

**Table 1 cells-11-00836-t001:** Moment and intensity of the maximal expression levels of differentiation markers in human keratinocytes (relative to D0).

	Gene	*KRT10*	*IVL*	*LOR*	*TGM1*	*FLG*	*AQP9*	*CDSN*	*KLK7*	*BNIP3L*
**Groups of age**	Young	D6	D6	D9	D6	D9	D9	D9	D9	D6
		244.7	38.7	984	24.9	201.4	127.7	576.8	215.8	34.7
	Adult	D9	D6	D6	D6	D9	D6	D6	D9	D9
		228.5	39.8	836	31.8	232.8	48.2	167.7	177.7	30.9
	Aged	D9	D9	D9	D6	D9	D9	D9	D9	D9
		52.4	25	189.5	25.2	139.7	25.2	74.1	35.8	8.9

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
