# Peer review of "MiR-30a-5p Alters Epidermal Terminal Differentiation during Aging by Regulating BNIP3L/NIX-Dependent Mitophagy"

_cells, 2022, doi:10.3390/cells11050836_

Round 1

Reviewer 1 Report

I have no further concern.

Author Response

We thank the reviewer for the constructive comments during the review process.

Reviewer 2 Report

In this manuscript by Chevalier et al entitled “MiR-30a-5p alters epidermal terminal differentiation during aging by regulating BNIP3L/NIX-dependent mitophagy”, the authors present the findings of the important role of miR-30a-5p in skin aging. The authors demonstrated that BNIP3L/NIX was a direct target of miR-30a. The authors also indicated that the decreased expression of BNIP3L and differentiation markers, together with a retention of mitochondria in aged keratinocytes. Therefore, the authors claimed that miR-30a in aged keratinocytes inhibit terminal differentiation through inhibition of BNIP3L-induced mitophagy. The manuscript are well written, and the results are presented by illustrative figures and graphs. Additionally, the results are supported by experimental data and correctly evaluated and properly discussed.

Author Response

We thank the reviewer for the supportive evaluation of our work.

Reviewer 3 Report

Comments on Manuscript submitted to Cells ID Number: cells-1563567

MiR-30a-5p alters epidermal terminal differentiation during aging by regulating BNIP3L/NIX-dependent mitophagy

 Comments to manuscript:

Novelty:

This work is the first to show that miR-30a affects mitophagy and skin aging through targeting BNIP3L.

However, Yan Guo et al showed that BNIP3L is the target of miR-30e-5p in Am J Physiol Renal Physiol. 2017 Apr 1;312(4): F589-F598. doi: 10.1152/ajprenal.00486.2016.  

In addition, Bin Du et al showed that BNIP3L is the target of miR-30c-5p in Cell Death Dis. 2017 Aug 10;8(8):e2987. doi: 10.1038/cddis.2017.377

Both have the same seed as miR-30a-5p therefore it is not surprising that miR-30a-5p also targets BNIP3L.

Although these are different members of the miR-30 family, it was shown that sometimes miRNAs with the same seed sequence don’t target the same mRNA. Moreover, the above works are not in the skin. The authors should address these issues in the discussion and at list quote these works

Comment

1) The authors show that mimic miR-30a-5p decreases BNIP3L mRNA expression 1B. With luciferase reporter plasmid that contains the 3’UTR of BNIP3L, they show very nicely that miR-30a-5p and not -3p reduce the luciferase activity. In addition, mutating the miR-30a-5p putative binding sites to thus 3’UTR diminish this reparation.

Although the authors show quantitative assay, (using ImageJ software) based on an immunofluorescence assay of BNIP3L which is a very visible assay but less quantitative. I think that a conclusively prove that BNIP3L is a biochemical target of miR-30a-5p they should add a Western blot experiment which mimic miR-30a-5p transfect to primary keratinocytes and analyzed BNIP3L expression.

 2) the author examined the expression of BNIP3L in human skin biopsies of different ages.

Were the skin biopsies were taken from the same body part?

3) In figure 4 and table 1 what is the expression of miR-30a-5p in these samples? Is it correlated to the expression of BNIP3L?

The paper should be published in the cells after the changes requested 

Conclusion

The manuscript should be published in the Cells Journal after the changes requested  

Author Response

We thank the reviewer for the constructive comments. We provided a point-by-point answers in the attached Word file.

Reviewer 4 Report

The specifics of the Real Time PCR cycles that have been performed for the different genes considered by the authors should be indicated in the text or in the supplementary data file. Were the cycles all the same? were the same annealing temperatures and melting curves set for all tested genes?
- did the authors not consider western blot to validate the data obtained with PCR?
- of lesser importance, in fig. 3 to make the arrangement of the images more linear it would be advisable to change the order of presentation of the first three: 35y, 38y and 42y

Author Response

We thank the reviewer for the constructive comments. We provided a point-by-point answers in the attached Word file.

This manuscript is a resubmission of an earlier submission. The following is a list of the peer review reports and author responses from that submission.

Round 1

Reviewer 1 Report

The paper is well written, objectives are clear and results convincing.

Minor points:

Figure 2: A panel showing the so-called "control of control" showing BNIP3L expression in human epidermis reconstructed using untransfected primary human keratinocytes is missing.

Figures 2 and 3: Indicating for example "donor 1", "donor 2", "donor 3" or the age of the donor, or something equivalent, above the 3 sets of images corresponding to 3 different donors, will help the reader to understand immediately the origin of the images.

Reviewer 2 Report

In this manuscript by Chevalier et al entitled “MiR-30a-5p alters epidermal terminal differentiation during aging by regulating BNIP3L/NIX-dependent mitophagy”, the authors present the findings of the important role of miR-30a-5p in skin aging. The authors demonstrated that BNIP3L/NIX was a direct target of miR-30a. The authors also indicated that the decreased expression of BNIP3L and differentiation markers, together with a retention of mitochondria in aged keratinocytes. Therefore, the authors claimed that miR-30a inhibit terminal differentiation of epidermal keratinocytes through inhibition of BNIP3L-induced mitophagy. However, the enthusiasm for this manuscript is limited by several major concerns, including the descriptive nature of the work and the lack of adequate support to the conclusions of the study by the data presented.

Major points:

  1. In figure 2, overexpression of miR-30a in reconstructed human epidermis resulted in the ~80% reduction of BNIP3L in the granular layers. However, in figure 1B, miR-30a inhibit BNIP3L expression levels by only 36%. As previous reports by the authors demonstrated that miR-30a overexpression in reconstituted human epidermis resulted in the strong inhibition of terminal differentiation, BNIP3L reduction in miR-30a overexpressed reconstituted human epidermis might be the secondary effect rather than direct repression of BNIP3L by miR-30a. The authors should clarify this point.

  1. Figure 5: The authors indicated that mitochondrial metabolic activity of aged keratinocytes was decreased. However, no relationship between this decrease in mitochondrial metabolic activity and BNIP3L-induced mitophagy or miR-30a expression in aged cells have been presented.